# The Supratrochlear Artery Sign—A New Piece in the Puzzle of Cerebral Vasospasm

**DOI:** 10.3390/diagnostics12092185

**Published:** 2022-09-09

**Authors:** Cindy Richter, Robert Werdehausen, Jennifer Jentzsch, Dirk Lindner, Thilo Gerhards, Torsten Hantel, Khaled Gaber, Stefan Schob, Dorothee Saur, Ulf Quäschling, Karl-Titus Hoffmann, Svitlana Ziganshyna, Dirk Halama

**Affiliations:** 1Department of Neuroradiology, University of Leipzig Medical Center, 04103 Leipzig, Germany; 2Department of Anesthesiology and Intensive Care Medicine, University of Leipzig Medical Center, 04103 Leipzig, Germany; 3Department of Neurosurgery, University of Leipzig Medical Center, 04103 Leipzig, Germany; 4Department of Radiology and Neuroradiology, Sana Hospital Borna, 04552 Borna, Germany; 5Department of Radiology, Halle University Hospital, 06120 Halle, Germany; 6Department of Neurology, University of Leipzig Medical Center, 04103 Leipzig, Germany; 7Department of Radiology, Kantonsspital Baselland, 4410 Liestal, Switzerland; 8Transplant Coordinator Unit, University of Leipzig Medical Center, 04103 Leipzig, Germany; 9Department of Oral and Maxillofacial Surgery, University of Leipzig Medical Center, 04103 Leipzig, Germany

**Keywords:** endovascular procedures, subarachnoid hemorrhage, supratrochlear artery, delayed cerebral ischemia, cerebral vasospasm

## Abstract

Background: Cerebral vasospasm (CVS) after subarachnoid hemorrhage (SAH) has been extensively investigated, but the impact of collateralization remains unclear. We investigated the predictive value of collateral activation for delayed cerebral ischemia (DCI)-related infarctions and functional outcome. Methods: Data from 43 patients with CVS (January 2014 to August 2021) were evaluated for the angiographic presence of leptomeningeal and ophthalmic collaterals (anterior falcine artery (AFA), supratrochlear artery (STA), dorsal nasal artery (DNA)) on internal carotid artery angiograms. Vasospasm-related infarction and the modified Rankin Scale (mRS) score after six months were chosen as the endpoints. Results: 77% of the patients suffered from DCI-related infarctions. In 233 angiograms (at hospitalization, before spasmolysis, after six months), positive vessel signs were observed in 31 patients for STA, 35 for DNA, and 31 for AFA. The STA sign had the highest positive (84.6%) and negative (85.7%) predictive value for unfavorable outcome (mRS 4–6) in patients aged ≥50 years. DNA and AFA signs were not meaningful predictors for either endpoint. Leptomeningeal collaterals showed a positive Pearson’s correlation with the STA sign in 87.5% (*p* = 0.038) without providing any prediction for either endpoint. Conclusions: The STA sign is associated with clinical outcome in patients with CVS after SAH aged ≥50 years, and was correlated with the occurrence of leptomeningeal collaterals.

## 1. Introduction

Cerebral vasospasm after aneurysmal subarachnoid hemorrhage (SAH) provokes acute hypoperfusion of brain parenchyma, demanding a fast adaption of collateral circulation. This self-limiting narrowing (vasospasm) of the cerebral vessels has a typical onset of three to five days after hemorrhage, with a maximum occurring usually at five to fourteen days, and gradual resolution over two to four weeks. The pathophysiological mechanism still remains unclear [1].

Narrowing of the terminal portion of the internal carotid artery (ICA) even disturbs the primary pathways of collateral circulation by the communicating arteries originating from the affected ICA segment [2]. Ophthalmic, leptomeningeal, and external carotid artery (ECA) collaterals can develop as secondary pathways in response to chronic hypoperfusion [3]. Leptomeningeal collaterals are small pial anastomoses connecting the territories of the middle cerebral artery (MCA), anterior cerebral artery (ACA), and the posterior cerebral artery (PCA). There are numerous anastomotic channels between branches of the ECA and the ICA, e.g., the dural arteriolar anastomoses: the anterior meningeal artery with the anterior falcine artery (AFA, anterior ethmoidal artery) and the pericallosal artery (branch of the ACA), the middle meningeal artery (branch of the maxillary artery), and the inferolateral trunk (branch of the ICA) or meningohypophyseal trunk (branch of the ICA). Communications between the two ECAs can frequently occur via their maxillary branches, because of widespread anastomoses in the nose and palate. Additionally, anastomoses are involved between occipital arteries, superficial temporal arteries, and the thyroid arterial plexus [4]. Collaterals can be visualized by digital subtraction angiography (DSA), which is the gold standard because of its high temporal and spatial resolution.

There is a watershed (border zones of distinct perfusion) in the supraorbital and infraorbital region between the distal branches of the ICA and ECA (Figure 1). It has been long established that the detection of flow changes in the supraorbital artery (SOA) and supratrochlear artery (STA) indicate severe carotid artery obstruction [5]. The ophthalmic artery is usually supplied by the intracranial portion of the ICA. Its terminal branches—the STA, the dorsal nasal artery (DNA), and the SOA—anastomose with the superficial temporal artery and the angular termination of the facial artery. If the extracranial or infraophthalmic intracranial ICA is significantly stenosed, the resulting pressure gradient results in a shift of the watershed towards the intraorbital arteries, with a retrograde flow in the STA and SOA. Normal ophthalmic flow can be maintained if the obstruction is compensated by anterior or posterior communicating arteries [6]. Proximal occlusions are more effectively compensated by collateral arteries than distal arteries.

Vasospasm of the supraopthalmic ICA may have a reversed effect, shifting the watershed towards the extraorbital arteries. Under physiological conditions, the STA, the DNA, and AFA are not detectible on ICA angiograms. In cerebral vasospasm, we observed an enlargement of the terminal branches of the ophthalmic artery, becoming angiographically detectable as vessel signs of the STA, DNA, and AFA. To the best of our knowledge, our study group is the first to describe these vessel signs. Current literature provides only scarce evidence on the significance of watershed shifts and cerebral collateral circulation in patients with symptomatic vasospasms [7].

We aimed to investigate the effect of cerebral collateral activation in vasospasm on delayed cerebral ischemia (DCI)-related infarction and patient outcomes. Leptomeningeal collaterals and ophthalmic vessel signs were analyzed by DSA to evaluate their predictive value for functional patient outcomes and DCI-related infarctions beyond mere description. We hypothesize that severe, hemodynamically relevant vasospasm can be angiographically revealed by enlarged terminal branches of the ophthalmic artery (vessel signs of the STA, AFA, DNA), and that the extent of DCI-related infarctions may differ in patients with poor or abundant collaterals. Hence, intact collateral circulation might be associated with a favorable clinical outcome after six months.

## 2. Materials and Methods

### 2.1. Study Population

All consecutive patients (*n* = 392) presenting to our center from January 2014 to August 2021 with acute aneurysmal SAH were retrospectively identified in our radiology information system. Only patients with subsequent cerebral vasospasm who had undergone at least one intra-arterial spasmolysis were selected for this study (*n* = 46).

The local Ethics Committee of our Medical Faculty approved this study (reference number: 412/19-ek) and all patients gave their informed consent. Inclusion criteria were (1) an aneurysmatic SAH verified by computer tomography (CT), magnetic resonance imaging (MRI), or lumbar puncture, and (2) pharmacological intra-arterial spasmolysis. Patients with mycotic or traumatic pseudoaneurysms, aneurysms associated with arteriovenous malformations or dissections were excluded.

The retrieved clinical information included demographic data (age, sex) and the initial SAH-related patient characteristics (Hunt and Hess scale, type and time-point of treatment of the vascular lesion). One author (CR), who was blinded to patients outcomes, retrospectively graded each non-contrast head CT on admission for SAH using the Fisher scale [8]. After six months, all patients were routinely summoned to our clinic to evaluate the functional outcome. The modified Rankin Scale (mRS) [9] was subsequently assessed based on the medical reports.

### 2.2. Infarction Pattern

The available non-contrast CT scans at hospitalization, after aneurysm treatment, at the onset of vasospasm before endovascular treatment, and after the vasospasm phase before discharge were analyzed for signs of infarction. The patterns of DCI-related infarction attributable to cerebral vasospasm can be divided into microangiopathic (lacunar) and macroangiopathic (territorial) lesions. We differentiated five size-dependent categories: (0) no infarctions, (1) lacunar infarctions ≤15 mm in size, (2) small territorial infarctions involving less or equal to a third of the respective vessel territory, (3) large territorial infarctions involving more than a third of the respective vessel territory, and (4) subtotal hemispheric infarctions. In cases of more than one lesion, the largest infarction determined the category. DCI was defined as the occurrence of DCI-related infarctions.

### 2.3. Clinical Management

All patients were monitored in our neurointensive care unit. Nimodipine was given from the day of admission, orally (6 × 60 mg/day) or via gastric tube. Mean arterial blood pressure was always sustained above 80 mmHg to support a cerebral perfusion pressure (CPP) over 70 mmHg.

Monitoring included daily transcranial Doppler investigation (TCD) of cerebral arteries. Cerebral vasospasm was suspected based on an altered level of consciousness or new neurological impairment (*n* = 16) or increased flow velocity in TCD above 180 cm/s in MCA or doubling of flow velocity within 24 h (*n* = 20). CT was performed in 42 patients and new territorial infarcts were detected in 5 patients. CT perfusion analysis was additionally performed in cases with ambiguous findings (*n* = 28). Six patients revealed a perfusion deficit leading to intra-arterial spasmolysis. Diagnostic DSA was performed if clinical worsening persisted, but TCD and CT-perfusion investigations did not reveal cerebral vasospasm (*n* = 1).

### 2.4. Diagnostic Cerebral Angiography and Intra-Arterial Spasmolysis

Diagnostic DSA was performed using a biplane system (Axiom Artis, Siemens, Erlangen, Germany or AlluraClarity, Philips Healthcare, Best, The Netherlands) or a monoplane system (Innova 4100; GE Healthcare, Waukesha, WI, USA). Iopromid (60–120 mL, containing 300 mg iodine per mL) was used as the contrast agent.

Every patient who underwent intra-arterial spasmolysis primarily received a diagnostic DSA to assess the severity of angiographic cerebral vasospasm. DSA was performed according to vascular regions with suspected cerebral vasospasm. A comparison of the diagnostic DSA and DSA on admission, if applicable, was performed. Therefore, not all segments were necessarily examined in every patient at every time point, especially not the vertebrobasilar region.

We regularly performed intra-arterial administration of spasmolytics over the entire study period, first using nimodipine, later using milrinone, or a combination of both. Nitro-glycerine and/or alprostadil were added as expanded access if the effect of the primarily given substances was insufficient.

### 2.5. Vasospasm Classification

All digital subtraction angiograms were graded with the visual cerebral vasospasm classification according to Merkel et al. [10]. Four grades of cerebral vasospasm (CVSG 0–3) are defined:CVSG 0—all intracranial vessels show a physiological shape;CVSG 1—Vasospasm affects the A2, A1, and M2 segments;CVSG 2—Vasospasm expands to the M1 and terminal segment of the ICA;CVSG 3—Severe reduction of the intradural internal carotid artery with filiform A1 and M1 segments, sometimes with a ghost-like fading appearance (ghost sign) [10].

### 2.6. Vessel Signs

The following vessel signs on DSA are not established. Naturally, these described vessels should not be present on an ICA angiogram if there is a balanced blood flow between the ECA and ICA. The angiographic presentability of the following branches of the ophthalmic artery were evaluated as vessel signs (Figure 2):The supratrochlear artery (STA) is one of the terminal branches of the ophthalmic artery. It pierces the septum orbitale at the medial corner of the orbit, superiorly. This artery supplies the skin, muscles, and the periosteum of the forehead. It anastomoses with the temporal artery and contralateral STA. It can be identified on an ICA angiogram in posterior–anterior (p.a.) and lateral projections. We considered the STA sign positive if it crossed the eyebrow in lateral projections. In p.a. projections, it had to be present at the paramedial zone (in the arterial or parenchymal phase) in parallel course with the A2 segment of the ACA. It can be easily mistaken for the supraorbital artery. The supraorbital artery is the smaller one, which runs laterally, and is often overlayed in p.a. projections. In lateral projections, it crosses the protruding part of the eyebrow (Figure 2D).The dorsal nasal artery (DNA) is the second terminal branch of the ophthalmic artery. It pierces the septum orbitale above the medial palpebral ligament to supply the skin on the roof of the nose and lacrimal sac region with the infraorbital artery. It anastomoses with the angular termination of the facial artery and the contralateral DNA. It can be identified with a parallel course to the nose on the ICA angiogram in p.a. projection. We considered the DNA sign positive if there was an artery (in arterial or parenchymal phase) in projection to the nose detectable.T-sign: The STA and the DNA branch from the same vessel origin forming a twisted “T” if both vessels are present on the ICA angiogram.The anterior falcine artery (AFA) is the meningeal branch of the anterior ethmoidal artery. It supplies the anterior portion of the falx cerebri with blood. The anterior ethmoidal artery arises from the third portion of the ophthalmic artery beneath the superior oblique muscle. It passes through the anterior ethmoidal canal and perforates the cribriform plate. On ICA angiograms, it is only detectable in lateral projections. We considered the AFA sign positive if there was a vessel on the inner side of the skullcap detectable (in the arterial or parenchymal phase).

### 2.7. Statistical Analysis

Descriptive statistical analyses were performed with SPSS version 27.0 (IBM Corporation; New York, NY, USA) and R version 4.0.1 (The R Foundation for Statistical Computing). Data were analyzed for Gaussian distribution with the Shapiro–Wilk test and using the Student’s *t*-test or Fisher’s test as appropriate. The Pearson’s coefficient was calculated for metric and the Spearman correlation coefficient for ordinal data. A two-tailed value of *p* < 0.05 was considered indicative of sufficient statistical significance.

## 3. Results

### 3.1. Data Collection

Forty-six patients (11.7%) experienced protracted cerebral vasospasm after SAH. Three patients with previous flow diverter implantation of the intracranial ICA were excluded from further analysis because of assumed artificial vessel diameter alteration. The remaining 43 patients were aged 29 to 79 years (mean: 49.6 years) and were predominantly female (2:1) (Table 1). Forty-two patients underwent endovascular treatment or clipping (coiling *n* = 33, clipping *n* = 5, coiling and clipping *n* = 3, coiling and flow diversion *n* = 1) within 48 h after SAH when immediately admitted to the hospital or at the time of hospitalization. In one case, an MCA aneurysm was treated with initial subtotal coiling followed by additional flow diversion in the recovery phase (“plug and pipe” procedure). In another case, no hemorrhage source was found.

### 3.2. Overall Outcome

Patients suffering from DCI-related infarction (77%; *n* = 33) had a favorable functional outcome (mRS 0–3) in 30.2% of cases (*n* = 13), while 27.4% (*n* = 20) had an unfavorable outcome (mRS 4–6) (Table 1). Patients without DCI-related infarction (23%; *n* = 10) had a favorable functional outcome (mRS 0–3) in 60% (*n* = 6), while the remaining fraction of 40% (*n* = 4) had an unfavorable outcome.

Ruptured aneurysms of the vertebrobasilar circulation were associated with an unfavorable outcome of mRS 5–6 (7%; *n* = 3). All fatal outcomes (mRS 6, 12%; *n* = 5) were associated with DCI-related infarctions. However, an unfavorable outcome was not always related to DCI-related infarctions. Overall, twenty-four patients suffered from an intracerebral hemorrhage (ICH) or/and severe complications like systemic inflammatory response syndrome, sepsis, pneumonia, or renal failure, that worsened the functional outcome (mRS 0–3 21%; *n* = 5, mRS 4–6 79%; *n* = 19). ICHs and non-DCI-related severe complications were summarized in the following as major complications.

### 3.3. Vessel Signs

Two neurointerventionalists (CR, TG), blinded to additional patient data, independently reviewed 233 DSA investigations of all 43 included patients. They compared angiograms on admission (*n* = 33) and the follow-up examination after six months (*n* = 21) with angiograms before intra-arterial spasmolysis (*n* = 178) to assess cerebral vasospasm grades for each examination. Vessel signs were only examined on admission and before intra-arterial spasmolysis. In cases with protracted cerebral vasospasm the treatment was repeated up to 14 times resulting in a large number of interventions. The results are presented in Table 2. For radiation protection, the forehead was not always depicted completely in lateral projections (STA *n* = 24 (55.8%), AFA *n* = 35 (81.4%)). Therefore, the STA sign could not be evaluated in p.a. projection in 19 cases. The AFA sign could not be assessed in eight cases (drop out). Five patients died and twelve patients did not undergo follow-up DSA after 6 months because of poor clinical outcomes (mRS 4–5). Follow-up DSA was not performed after clipping procedures.

Four patients (9%) presented with a positive STA sign already at admission. One of these patients suffered from early angiographic vasospasm (EAVS). Positive DNA (81%; *n* = 35) and AFA (72%; *n* = 31) signs correlated less with progressive vasospasm grades already showing positive vessel signs at time of admission (DNA: 30.2%, *n* = 13; AFA: 25%, *n* = 9). In EAVS, most patients (aged 40–68 years) revealed a negative vessel sign: STA (90%, *n*= 9, mean: 53.1 years), DNA (80%, *n* = 8, mean: 52.3 years) and AFA (77.8%, *n* = 7, mean: 52.3 years).

The mean patient age for a positive STA sign was 47.9 ± 12.0 years compared to patients aged 54.0 ± 11.1 years with negative STA sign. A positive DNA sign was found in 72.1% (*n* = 31) of patients aged 48.6 ± 11.5 years, while no DNA sign was observed in 27.9% (*n* = 12) of patients aged 54.1 ± 13.8 years. The AFA sign was positive in 86.1% (*n* = 31) of patients aged 48.6 ± 10.4 years and absent in those aged 50.4 ± 18.1 years.

Vessel signs were compared to functional outcome (mRS scale) and DCI-related infarctions by means of Fisher’s test (Table 3). To assess age-dependent variations, three groups were used for calculations, all patients (*n* = 43), patients < 50 years (*n* = 23) and patients ≥ 50 years (*n* = 20). The functional outcome was divided into favorable (mRS 0–3) and unfavorable outcomes (mRS 4–6). The Fisher’s test indicated a significant *p*-value of 0.004 for the STA sign in patients ≥50 years for the patients’ functional outcome. The positive predictive value for a positive STA sign and an unfavorable outcome was 84.6%. The negative predictive value for a negative STA sign and favorable outcome was 85.7%. For patients younger than 50 years, the Fisher’s test revealed a non-significant *p*-value of 0.281 for the functional outcome, with a positive predictive value of 70% and a negative predictive value of 7.7%. There were no significant predictive values of the AFA and DNA signs.

The distribution of patients’ outcomes with or without a STA sign is presented in Figure 3. While calculations for assessing the predictive value of the STA sign were done with data from all included patients (upper panel), it became clear that major complications worsened patient outcomes with increasing age (middle panel). Considering only patients without major complications (lower panel), a favorable outcome was achieved in most cases, especially with negative STA signs.

With regard to cerebral infarctions, more patients with a negative STA sign (42%) were free from DCI-related infarctions compared to those with a positive STA sign (16%, *p* = 0.110, Table 3, Figure 4) without any significant prediction. For the AFA sign and DNA sign, no differences in cerebral infarctions were observed (Figure 4). Subtotal hemispheric infarctions were always associated with positive vessel signs. The five different categories of vasospasm-related infarctions (0 = none, 1 = lacunar infarction, 2 = small territorial infarctions, 4 = large territorial infarctions, 4 = subtotal hemispheric infarctions) were correlated with each vessel sign. The Spearman correlation coefficient did not show any significant correlation for STA, AFA, or DNA signs with the infarction pattern (STA: ρ = 0.198, *p*-value = 0.204, AFA: ρ = −0.026, *p*-value = 0.867, DNA ρ = −0.038, *p*-value = 0.809).

### 3.4. Leptomeningeal Collaterals

The activation of leptomeningeal collaterals between the MCA and ACA territory was assessed in p.a. projection. The varying ability of collateral supply is shown in Figure 5 and Figure 6. Two patients first had activated collateral supply, but later had collapsed collateral supply (Figure 6). The distinct outcomes with mRS values of 3 and 5, despite subtotal hemispheric infarction, were caused by the laterality of language ability.

In all cases with collateral supply, vasospasm had affected the ACA hardest, requiring leptomeningeal collaterals to take over the anterior and border zone supply. Only 37.2% (*n* = 16) of patients revealed a leptomeningeal collateral activation on angiograms (Table 1). Fourteen patients (87.5%) with leptomeningeal collateral activation showed a positive STA sign on angiograms with a positive correlation in Pearson’s correlation (Correlation coefficient: 0.318, *p* = 0.038). Overall, Fisher’s test indicated no significance for the prediction of outcome (*p* = 1) or DCI (*p* = 0.719) in the context of leptomeningeal activation.

The vertebrobasilar system was inconsistently depicted. Eight patients never obtained angiographic imaging of the vertebral artery before intra-arterial spasmolysis. The collateral circulation of the vertebrobasilar territory was not considered for this evaluation. Eight of the remaining thirty-five patients (23%) revealed vertebrobasilar vasospasm with an unfavorable median functional outcome of mRS 4 (Table 1).

## 4. Discussion

Although the development and prevalence of cerebral vasospasm have been extensively investigated, little is known about the extent of collateralization and its predictive value for the clinical outcome [7]. In ischemic stroke, the activation of collateral pathways has fundamental prognostic importance [11,12]. In our study, 76.7% (*n* = 33) of patients suffered from DCI-related infarctions. Stenoses of the anterior circulation in cerebral vasospasm are mostly located distal to the ophthalmic artery origin, similar to moyamoya disease. The development of collaterals supports the maintenance of minimal perfusion to the cerebral parenchyma, particularly of the ACA territories [13,14]. A detailed analysis of the potential role of the ophthalmic artery in supplying the ACA and MCA territories is lacking in the literature.

In cerebral vasospasm, acute vessel narrowing may cause acute hypoperfusion. Neurons will only survive long enough to be rescued with reperfusion therapies if there is sufficient collateral flow. The vessel signs we evaluated, indicating the realignment of blood flow under critical conditions, did not always become positive, even if patients suffered from high-grade vasospasm (CVSG 3). Rosenkranz et al. [15] reported that proximal ICA stenosis of less than 80% is not mandatorily associated with activation of primary collateral vessels. The extent of collateral flow is highly variable between individuals. Consequently, the resulting DCI-related infarctions in cerebral vasospasm do not solely depend on the severity of visible vasospasm. However, there are some differences between cerebral vasospasm, steno-occlusive diseases as moyamoya, and degenerative stenosis of intracranial vessels. Stenosis of vessels often develops over a long period, which provides the opportunity to obtain collateral circuits over a long time. Proximal ICA occlusion leaves the circle of Willis open as the primary collateral route. Cerebral vasospasm also affects vessels beyond the circle of Willis, depending on secondary collateral routes. Leptomeningeal collaterals represent potential endogenous bypass vessels, capable of maintaining blood flow to brain regions that would be injured during acute vasospasm. There is robust evidence from imaging studies among patients with acute ischemic stroke that these collaterals at baseline exhibit substantial variability [3].

Only 32.7% of the patients in our study developed angiographically visible leptomeningeal collaterals for ACA territory supply. However, in our analysis, these collaterals had no significant predictive value for patients’ outcome or DCI-related infarctions. In accordance with the literature [16], patients with fewer or collapsed leptomeningeal collateral vessels had worse outcomes, excluding other severe complications. Subtotal hemispheric infarctions closely followed collapsed collaterals.

Some case series reported about extracranial-intracranial bypass procedures for treating vasospasm after SAH if aggressive medical management fails [17]. Batjer et al. [18] reported regarding eleven patients. In six of eight cases with predominantly proximal vasospasm involving the ICA and M1 trunks, neurological deficits improved or resolved after bypass procedure within 24 h. Both patients with proximal and distal vasospasm maintained their preoperative neurological status with tolerated lower mean arterial blood pressure. One patient with isolated distal MCA vasospasm failed to respond. This treatment did not gain acceptance because of the association with significant operative mortality and morbidity [18,19].

The most common secondary collateral is the connection through the ophthalmic artery. The DNA and the STA are the terminal branches of the ophthalmic artery, with retrograde flow in proximal ICA occlusion and with enhanced antegrade flow in ICA occlusion distal to the ophthalmic origin. Likewise, the AFA is a meningeal branch of the anterior ethmoidal artery arising from the ophthalmic artery. If other extra-cerebral arteries are compromised (e.g., the MMA), it extends beyond the anterior frontal convexity. Abnormal enlargement of the AFA was seen in pathologic conditions that involve the falx or adjacent dura: meningiomas, arteriovenous malformations, Paget’s disease, lymphoma, chronic meningitis, and metastatic neoplasms [20]. The communications between meningeal arteries and cortical arteries were described by Mount and Taveras [21]. The ACA may communicate with leptomeningeal branches of the pericallosal and the callosomarginal arteries in stenosis or occlusions of the ICA distal to the ophthalmic artery [20].

The STA sign revealed a significant positive and negative predictive value for patient’s outcome in the group ≥50 years. Patients at risk for unfavorable outcomes (mRS 4–6) were aged <50 years, suffered from high-grade vasospasm, and showed a positive STA sign, or were aged ≥50 years and showed a positive STA sign with indifferent vasospasm grades. The STA sign indicates self-sufficient older patients at risk for unfavorable outcome who may benefit from intensified care and monitoring after SAH. Patients suffering from severe SAH are frequently in a poor neurological condition, and the diagnosis of DCI, according to Vergouwen et al. [22], is often challenging, with a broad inter-rater variance [23]. Here, the STA sign might be helpful diagnostic tool.

The STA is easily accessible for extracranial color-coded duplex sonography. In this method, a positive STA sign can also be detected with bedside monitoring, showing increase antegrade flow velocities for the affected side (Figure 7). Especially in patients aged ≥50 years, the transtemporal color-coded duplex sonography often fails due to a missing ultrasound window. This finding needs to be further evaluated. It represents a low-cost and non-invasive method to obtain hemodynamic information in real-time [5], even in patients without a temporal ultrasound window.

Compared to the STA sign, DNA and AFA signs were more often already present in angiograms on admission, and had no predictive value with regard to patient outcomes or DCI-related infarctions. In contrast to leptomeningeal collaterals, dural arteriolar anastomoses can only contribute a small amount of blood due to their small, less flexible diameters. Subtotal hemispheric infarction was always related to positive vessel signs of all three arteries. Additionally, younger patients more often displayed positive vessel signs without high-grade vasospasm, as well as better outcome. Moftakhar et al. [24] examined 37 children with aneurysmal SAH. Children have a relatively high incidence of angiographically detectable moderate-to-severe vasospasm, but rarely develop symptomatic cerebral vasospasm [25]. Unlike adults, children usually have good long-term outcomes, probably because of better cerebral collateral circulation. The increased stiffness of the cerebral vasculature associated with advanced age may explain the lower prevalence of cerebral vasospasm in the population of older patients. Aging leads to collagen accumulation in the vessel wall, with intimal thickening and fibrosis. These changes, and the accumulation of atherosclerotic plaque, decrease vessel distensibility. Vessel reactivity to mediators of vasoconstriction may also decrease with age [24].

Probably because of the small number of analyzed patients, we identified no significant predictive value of a positive STA sign for DCI-related infarction. In the past decades, evidence has shown that cerebral vasospasm and DCI should be considered as two separate entities. DCI is more complex than simply being a result of arterial narrowing. A multifactorial process involving blood-brain-barrier disruption, microthrombosis, cortical spreading depolarization, and loss of cerebral autoregulation leads to DCI [23].

### Limitations

This retrospective study with a focus on imaging findings has several limitations. First of all, the small, inhomogeneous sample size of the group does not allow for rigorous statistical assessment. Another limitation is the retrospective nature of this study and the lack of an independent dataset for evaluating the predictive value of collateralization for DCI-related infarction and functional outcome. CT scans and DSA were only performed when clinically indicated. This might have led to a detection bias favoring patients with poor clinical outcomes. The EAVS and the vertebrobasilar vasospasm could only be assessed for a small group of patients.

## 5. Conclusions

Of several vascular signs examined by DSA, the STA sign proved to be an indicator of the vulnerability to cerebral vasospasm after SAH. A positive STA sign was associated with an unfavorable outcome for patients ≥50 years, while a negative sign was associated with a favorable functional outcome in this patient group.

## Figures and Tables

**Figure 1 diagnostics-12-02185-f001:**
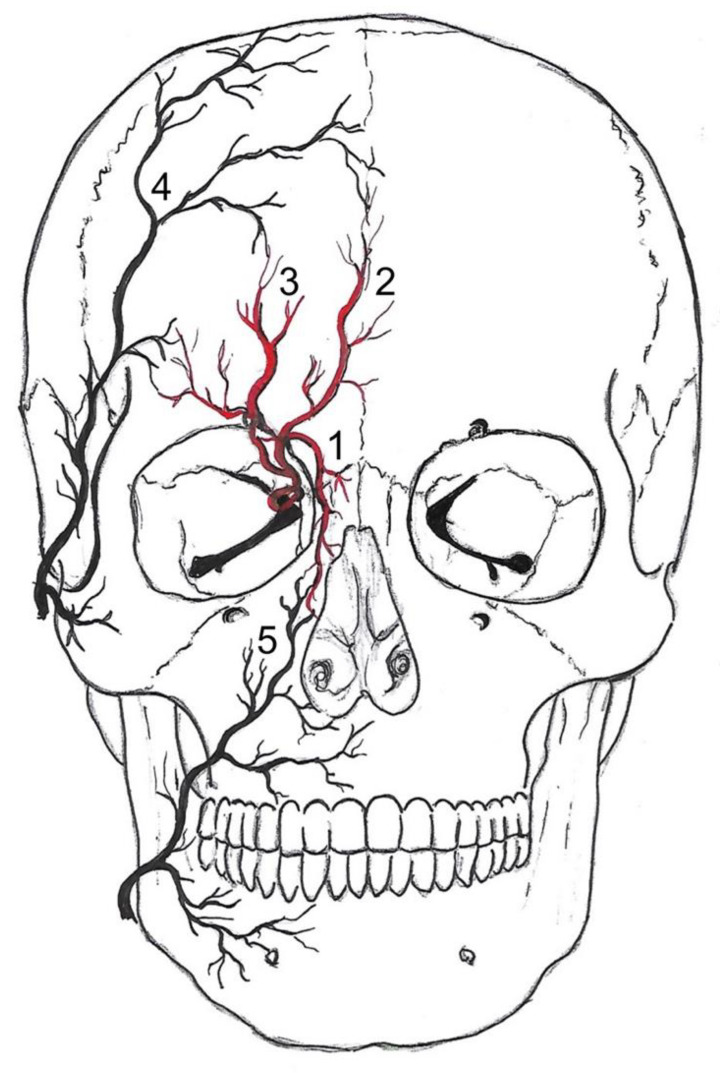
Anastomoses of the internal and external carotid arteries. Vessels drawn in red (1–3) belong to the internal carotid drainage and vessels depicted in black (4, 5) belong to external carotid artery drainage. The dorsal nasal artery (1) and the supratrochlear artery (2) are the terminal branches of the ophthalmic artery—a branch of the internal carotid artery. The supraorbital artery (3) is a branch of the intraorbital ophthalmic artery and passes through the supraorbital foramen. It anastomoses within the scalp with the superficial temporal artery (4) and the supratrochlear artery. The supratrochlear artery pierces the orbital septum at the medial corner of the orbit superiorly. It anastomoses with the superficial temporal artery and the contralateral supratrochlear artery. The dorsal nasal artery pierces the septum orbitale above the medial palpebral ligament. It anastomoses with the angular termination of the facial artery (5) and the contralateral dorsal nasal artery.

**Figure 2 diagnostics-12-02185-f002:**
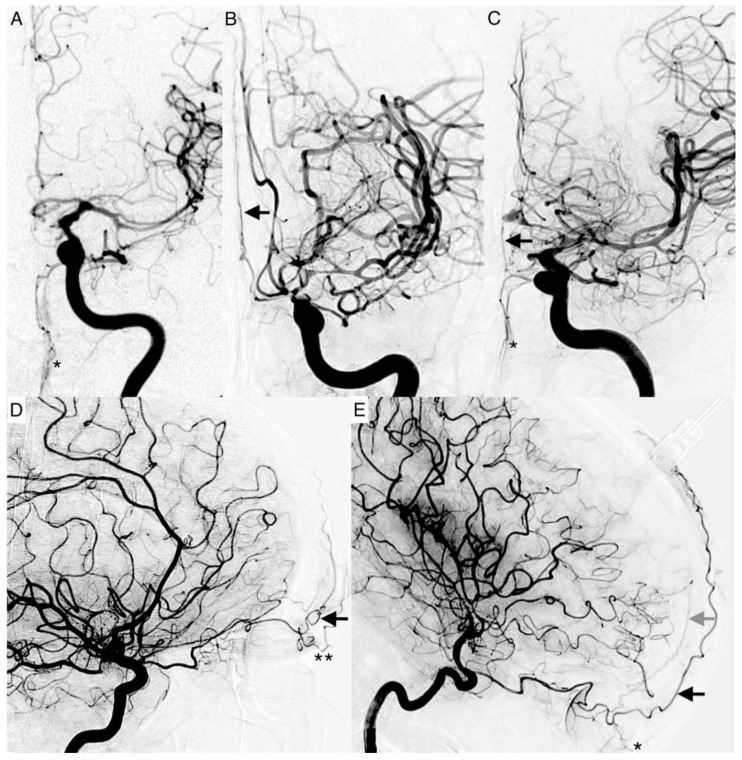
Angiograms of enlarged terminal branches of the ophthalmic artery. Posterior-anterior projections: (**A**) The dorsal nasal artery (DNA, single star) is shown as prominent terminal branch in moderate vasospasm. (**B**) The supratrochlear artery (STA, black arrow) enlarged in severe vasospasm with a ghost sign of the carotid T. The A2 segment of the anterior cerebral artery is not present. (**C**) Severe vasospasm with a positive twisted T-sign of the DNA (single star) and the STA (black arrow). Lateral projections: (**D**) Constitutional prominent STA (black arrow) and supraorbital artery (double stars) on admission. (**E**) Enlarged STA (black arrow), DNA (single star) and prominent anterior falx artery (AFA, gray arrow) in high-grade vasospasm with narrowing of the terminal internal carotid artery.

**Figure 3 diagnostics-12-02185-f003:**
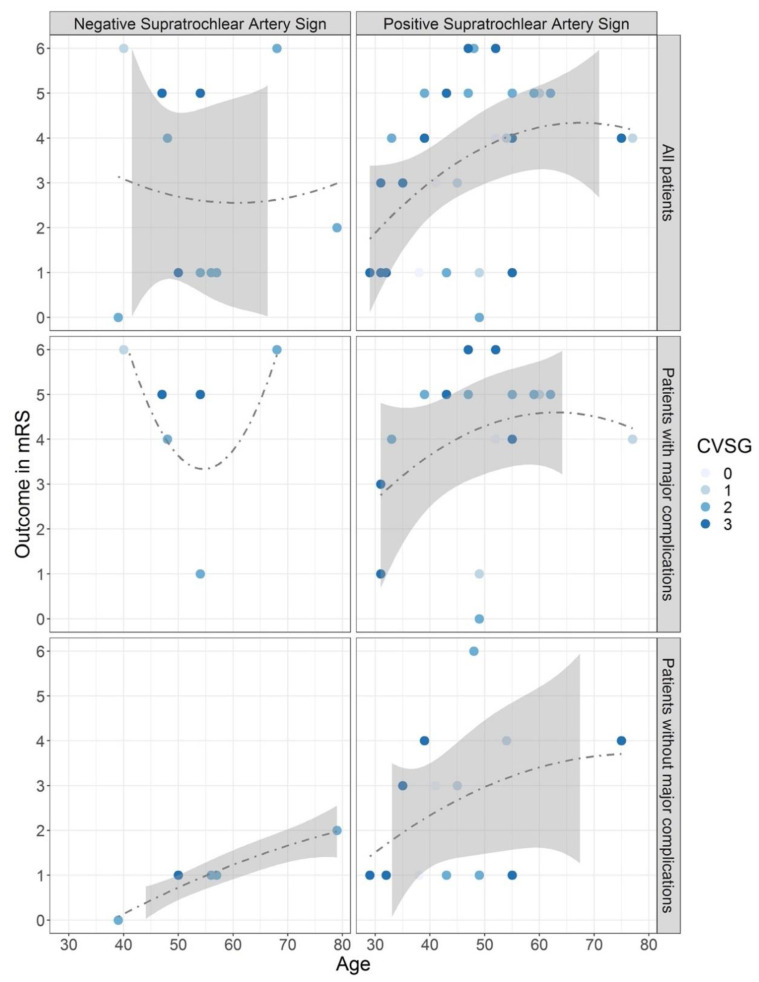
Supratrochlear artery sign with polynomial regression. The diagrams of the first line include all patients (*n* = 43). The graphs of the second line select patients suffering from major complications (*n* = 24) like intracerebral hemorrhage, systemic inflammatory response syndrome, sepsis, pneumonia, or renal failure, that worsened the functional outcome. The graphs of the third line depict patients without major complications (*n* = 19). The left and right columns show the rating for the negative and positive supratrochlear artery (STA) sign, respectively, according to patient’s age, outcome in modified Rankin Scale (mRS) score, cerebral vasospasm grade (CVSG) at the time-point of positive STA sign or highest CVSG for negative STA sign. The gray area displays the 95% confidence interval. Please note that the colored area is missing if it was out of the outcome range. The polynomial regression lines are graphed in grey. There is a significant correlation between unfavorable outcomes for positive supratrochlear artery signs with advancing age. The outcome is even aggravated with major complications (intracranial hemorrhage and/or severe co-morbidities). DNA and AFA signs did not prove to be meaningful predictors (not shown).

**Figure 4 diagnostics-12-02185-f004:**
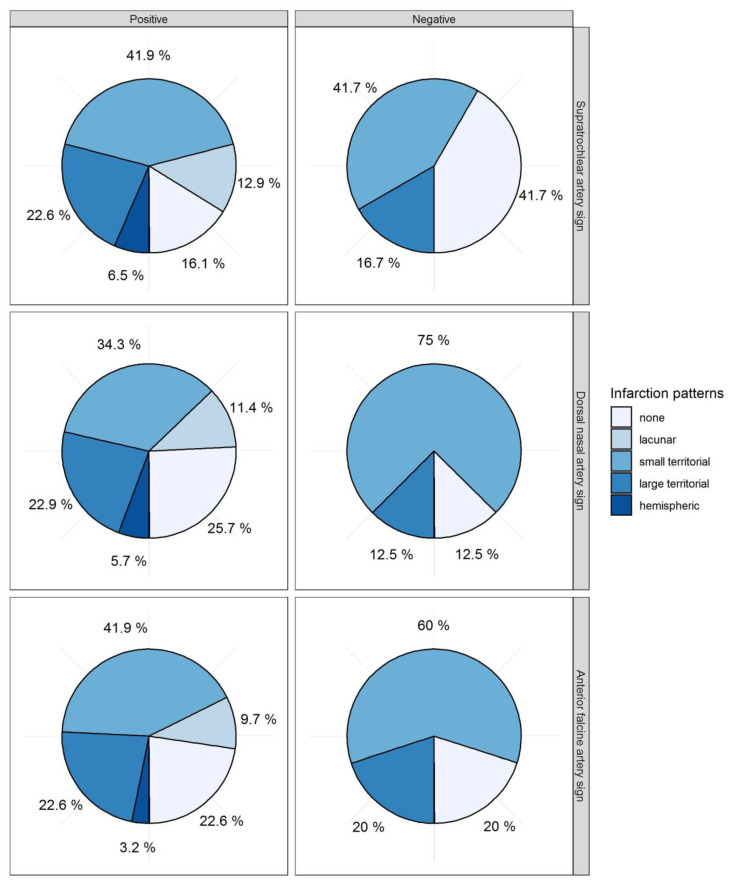
Infarction pattern. The distribution of infarction size is displayed in circle diagrams for each positive and negative vessel sign. Supratrochlear artery (STA) sign: positive *n* = 31 (72.1%), negative *n*= 12 (27.9%). Dorsal nasal artery (DNA) sign: positive *n* = 35 (81.4%), negative = 18.6%. Anterior falcine artery (AFA) sign: positive *n* = 31 (72.1%), negative *n* = 5 (11.6%). Eight cases were dropped out due to AFA sign, if the forehead was incompletely depicted (18.6%).

**Figure 5 diagnostics-12-02185-f005:**
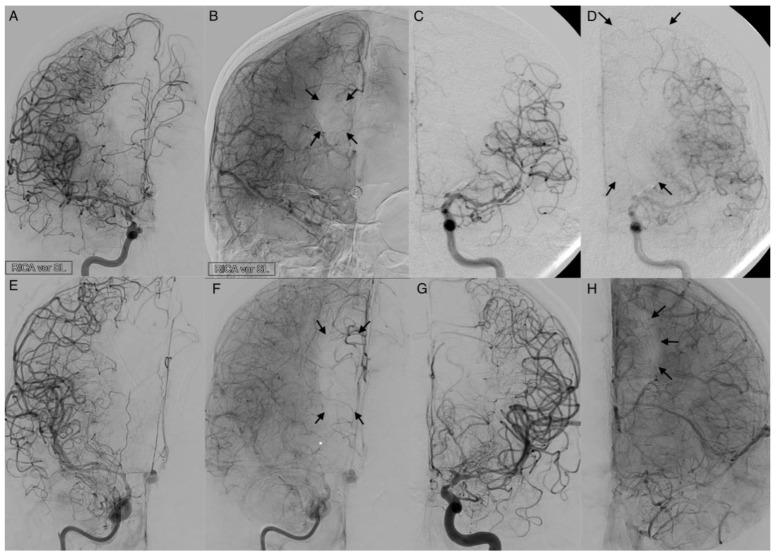
Activated leptomeningeal collaterals in anterior cerebral artery vasospasm. The angiograms show the extent of collateralization in four cases of severe cerebral vasospasm following aneurysmal subarachnoid hemorrhage. The arterial phase (**A**,**C**,**E**,**G**) reveals a strongly narrowed and decelerated anterior cerebral artery. In the parenchymal phase (**B**,**D**,**F**,**H**), activated leptomeningeal collaterals of the middle cerebral artery took over the retrograde supply of the decelerated parenchyma with varying extent of the remaining perfusion defect (arrows).

**Figure 6 diagnostics-12-02185-f006:**
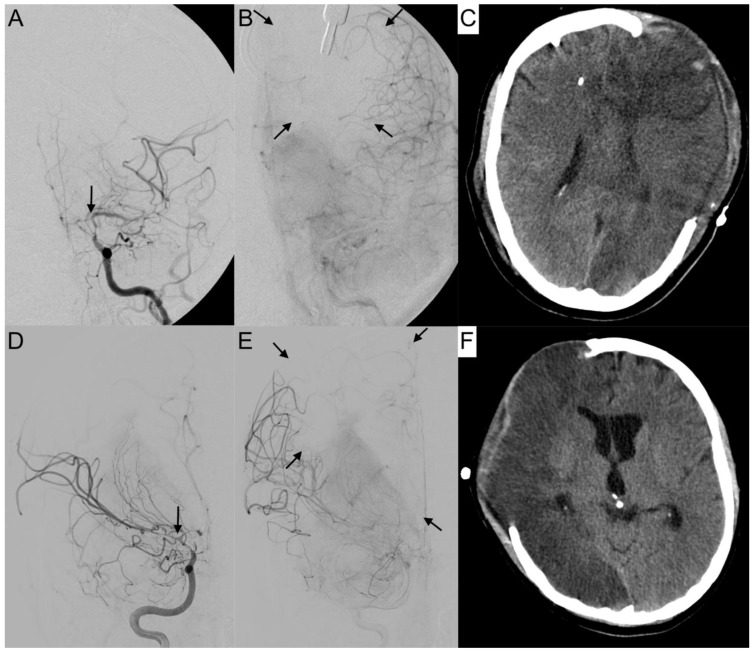
Collapsed collateral supply in severe vasospasm. Angiograms of the left (**A**,**B**) and right (**D**,**E**) internal carotid arteries (ICA) with severe vasospasm (grade 3) affecting the terminal portion of the ICA (**A**,**D**: arrows) are shown. In the parenchymal phase, the breakdown of collateral supply resulted in a large perfusion deficit (**B**,**E**: arrows), leading to subtotal hemispheric infarction (**C**,**F**) in computer tomography. Both patients revealed a distinct outcome with mRS 3 (**A**–**C**) and mRS 5 (**D**,**E**), depending also on the laterality of language ability.

**Figure 7 diagnostics-12-02185-f007:**
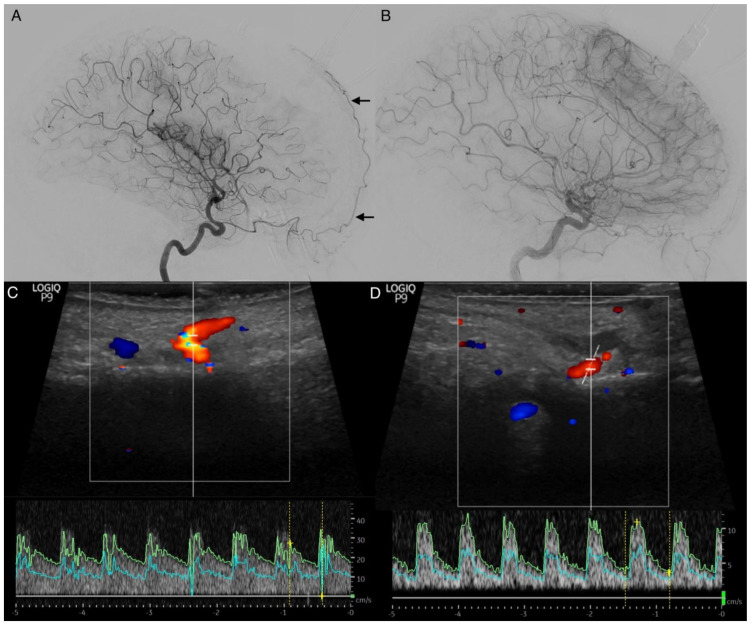
Doppler sonography of the supratrochlear artery in carotid artery vasospasm. (**A**,**B**): Lateral projections of the right (**A**) and left (**B**) anterior circulation in digital subtraction angiography. The right-sided angiogram shows the intradural internal carotid artery (ICA) vasospasm and an enlarged supratrochlear artery (STA, arrows). The right-sided angiogram shows the physiological appearance of the STA without intradural ICA vasospasm. (**C**,**D**): Transcutaneous ultrasound measurements of the right (**C**) and left (**D**) supratrochlear artery at the superior margin of the orbit. The flow velocity of the right supratrochlear artery in high-grade cerebral vasospasm was tripled compared to the counterside (mean flow velocity: right 12.3 cm/s, left 4.3 cm/s, systolic velocity: right 27.4 cm/s, left 11 cm/s; diastolic velocity: right 0 cm/s, left 3.7 cm/s; resistant index: right 1.00, left 0.66; pulsatility index: right 1.36; left 1.07).

**Table 1 diagnostics-12-02185-t001:** Clinical data.

Baseline Demographics	Subdivision		Outcome in mRS						
		*n*	0	1	2	3	4	5	6
Overall		43	2	13	1	4	8	10	5
Fisher Score	2	3	1	0	0	1	0	1	0
	3	10	0	5	1	0	2	1	1
	4	30	1	8	0	3	6	8	4
Hunt and Hess Score	1	5	1	0	0	1	2	1	0
	2	10	0	4	0	2	0	2	2
	3	13	1	4	1	0	3	3	1
	4	7	0	4	0	0	1	1	1
	5	7	0	1	0	1	2	2	1
Localization of aneurysm	AComA	0	1	4	0	3	2	1	0
	ACA	0	0	1	0	0	1	1	1
	MCA	0	1	4	0	1	3	4	2
	ICA/PComA	0	0	4	1	0	1	1	0
	VA/BA	0	0	0	0	0	0	1	2
	none	0	0	0	0	0	1	0	0
Clipping		5	0	1	0	0	1	1	2
Endovascular procedure		35	2	12	1	3	6	7	3
Combined therapy		3	0	0	0	1	0	2	0
**Vessel signs**									
Positive supratrochlear artery sign	at time of admission	4	0	2	1	1	0	0	0
(STA)	during vasospasm	31	1	8	0	4	7	8	3
Positive dorsal nasal artery sign (DNA)	at time of admission	13	0	6	0	1	3	2	1
	during vasospasm	35	1	11	0	4	7	8	4
Positive anterior falcine artery sign (AFA)	at time of admission	9	0	4	0	1	1	3	0
	during vasospasm	31	1	10	0	4	5	7	4
Leptomingeal collateral activation	non-activated	25	2	7	1	2	5	5	3
	activated	16	0	6	0	1	3	4	2
	collapsed	2	0	0	0	1	0	1	0
**Outcome-limiting factors**									
Intracerebral hemorrhage		13	0	3	0	1	3	5	1
Non-DCI-related infarctions		9	1	3	0	2	2	1	0
Additional severe co-morbidity		15	1	0	0	0	4	8	2
DCI-related infarctions		33	1	8	1	3	8	7	5
Lacunar infarction, size ≤ 15 mm		2	0	2	0	0	0	0	0
Small territorial infarction (≤1/3 of area at risk)		18	1	6	1	2	4	3	1
Large territorial infarction		9	0	0	0	0	2	3	4
Subtotal hemispheric infarction		2	0	0	0	1	0	1	0
Unilateral infarctions		23	1	6	1	1	7	5	2
Bilateral infarctions		10	0	2	2	2	1	2	3
Vertebrobasilar infarctions		8	0	0	1	1	2	1	3
Number of affected vessel territories	0	10	1	5	0	1	0	3	0
	1	14	1	6	0	0	4	2	1
	2	8	0	2	1	1	3	1	0
	3	4	0	0	0	1	0	3	0
	4	5	0	0	0	1	1	1	2
	6	2	0	0	0	0	0	0	2

Additional severe co-morbidities were systemic inflammatory response syndrome, sepsis, pneumonia, or renal failure, that worsened the functional outcome. *CVS* cerebral vasospasm, *DCI* delayed cerebral ischemia, *mRS* modified Rankin Scale, *ACA* anterior cerebral artery, *AComA* anterior communicating artery, *BA* basilar artery *ICA* internal carotid artery, *MCA* middle cerebral artery, *PComA* Posterior communicating artery, *VA* vertebral artery.

**Table 2 diagnostics-12-02185-t002:** Vessel signs in cerebral vasospasm.

**Positive Vessel Sign**	**CVSG on Admission**				
	**All**	**0**	**1**	**2**	**3**
Overall (%, *n*)	100 (43)	51.2 (22)	23.3 (10)	14.0 (6)	11.6 (5)
STA (%, *n*)	9.3 (4)	4.7 (2)	4.7 (2)	0.0 (0)	0.0 (0)
DNA (%, *n*)	30.2 (13)	20.9 (9)	7.0 (3)	0.0 (0)	2.3 (1)
AFA (%, *n*)	25.7 (9)	14.3 (5)	2.8 (3)	2.9 (1)	0.0 (0)
**Positive Vessel Sign**	**Early Angiographic Vasospasm in CVSG (<72 h)**				
	** *n* **	**0**	**1**	**2**	**3**
Overall (%, *n*)	100 (43)	0.0 (0)	14.0 (6)	4.7 (2)	4.7 (2)
STA (%, *n*)	2.3 (1)	0.0 (0)	2.3 (1)	0.0 (0)	0.0 (0)
DNA (%, *n*)	4.7 (2)	0.0 (0)	2.3 (1)	0.0 (0)	2.3 (1)
AFA (%, *n*)	8.6 (3)	0.0 (0)	8.6 (3)	0.0 (0)	0.0 (0)
**Positive Vessel Sign**	**Highest CVSG**				
	** *n* **	**0**	**1**	**2**	**3**
Overall (%, *n*)	100 (43)	4.7 (2)	7.0 (3)	44.2 (19)	44.2 (19)
STA (%, *n*)	72.1 (31)	4.7 (2)	4.7 (2)	27.9 (12)	34.9 (15)
DNA (%, *n*)	81.4 (35)	4.7 (2)	7.0 (3)	30.2 (13)	39.5 (17)
AFA (%, *n*)	72.1 (31)	5.7 (2)	5.7 (2)	37.1 (13)	40.0 (14)

Please note, that the AFA sign could not be evaluated for eight patients because the frontal area was not depicted in lateral projections. *CVSG* cerebral vasospasm grade, *STA* supratrochlear artery, *DNA* dorsal nasal artery, *AFA* anterior falcine artery.

**Table 3 diagnostics-12-02185-t003:** Vessel signs in cerebral vasospasm.

	Rating		Functional Outcome			Delayed Cerebral Ischemia		
	Positive Sign (%, *n*)	Negative Sign (%, *n*)	PPV (%)	NPV (%)	*p*-Value	PPV (%)	NPV (%)	*p*-Value
Overall	100 (43)							
Supratrochlear artery	72.1 (31)	27.9 (12)	78.3	35.0	0.497	78.8	50.0	0.110
Dorsal nasal artery	81.4 (35)	18.6 (8)	82.6	20.0	1.000	78.8	10.0	0.656
Anterior falcine artery	88.6 (31)	11.4 (4)	94.1	16.7	0.602	88.9	12.5	1.000
Patients < 50 years	53.5 (23)							
Supratrochlear artery	82.6 (19)	17.4 (4)	70.0	7.7	0.281	88.9	40.0	0.194
Dorsal nasal artery	82.6 (19)	17.4 (4)	80.0	15.4	1.000	83.3	25.0	1.000
Anterior falcine artery	85.7 (18)	14.3 (3)	88.9	16.7	1.000	87.5	20.0	1.000
Patients ≥ 50 years	46.5 (20)							
Supratrochlear artery	60.0(12)	40.0 (8)	84.6	85.7	0.004	66.0	60.0	0.347
Dorsal nasal artery	80.0 (16)	20.0 (4)	84.6	28.6	0.587	73.3	0.0	0.530
Anterior falcine artery	92.9 (13)	7.1 (1)	100.0	20.0	0.428	90.9	0.0	1.000

Please note, that the AFA sign could not be evaluated for eight patients because the frontal area was not depicted in lateral projections. *PPV* positive predictive value, *NPV* negative predictive value, Fisher’s test (*p*-values).

## Data Availability

The datasets generated and analyzed during the current study are available in the Zenado repository, https://www.zenodo.org (accessed on 25 June 2022), Digital Object Identifier Number: 10.5281/zenodo.6731348; further inquiries can be directed to the corresponding author.

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
