# Peer review of "The Supratrochlear Artery Sign—A New Piece in the Puzzle of Cerebral Vasospasm"

_diagnostics, 2022, doi:10.3390/diagnostics12092185_

Round 1

Reviewer 1 Report

This is an iteresting article, whch is nicely written and provides important information for neuro area. I recommend it for publication in its present form.

The introducition in concise and provides an insight in the research topic. The methods and results are clearly presented, angiograms are added, diagrams and the descriptions. The results are clear. 

The discussion is well written. I think that the tops is interesting and of merit for the clinical practice.  

Author Response

Response to Reviewer 1 Comments

This is an interesting article, which is nicely written and provides important information for neuro area. I recommend it for publication in its present form.

The introduction in concise and provides an insight in the research topic. The methods and results are clearly presented, angiograms are added, diagrams and the descriptions. The results are clear. 

The discussion is well written. I think that the tops is interesting and of merit for the clinical practice.  

Answer: We like to thank you very much for reviewing our manuscript and for your comments.

Reviewer 2 Report

This article studies the prediction of cerebral vasospasm (CVS) after subarachnoid hemorrhage (SAH). A retrospective study was conducted to explore the role of several DSA signs reflecting the activation of ophthalmic artery terminals after SAH as indicators of infarction and functional outcomes. The results showed that the STA sign was associated with clinical outcome in patients with CVS after SAH aged ≥50 years, and was associated with the occurrence of leptomeningeal collaterals. Moreover, the authors provide the predictive values of the presence or absence of the sign for the outcome.

The paper is innovative and draws conclusions based on the research data, and the research design is appropriate, which is adequately described in methods. The paper provides valuable DSA predictive signs for The occurrence of CVS after SAH, so that clinicians can more calmly cope with brain tissue ischemia or infarction after SAH.

This is a topic of interest to the researchers in the related areas but the paper needs some improvement before acceptance for publication. My detailed comments are as follows:

1. Abbreviations and acronyms should be defined within the text upon first appearance. Such as, in page 2, Introduction, line 56, ICA would be declared at first.

2. Please unify the terms in the paper, such as DCI, DCI-related infarction, and cerebral infarction.

3. In page 5, Materials and Methods 2.6, Please determine whether it is necessary to state how to distinguish STA from Supraorbital artery.

4. Please expand the significance of this study for clinical work and research in Discussion.

Therefore, our Overall Recommendation is: Accept after minor revision.

Author Response

Response to Reviewer 2 Comments

This article studies the prediction of cerebral vasospasm (CVS) after subarachnoid hemorrhage (SAH). A retrospective study was conducted to explore the role of several DSA signs reflecting the activation of ophthalmic artery terminals after SAH as indicators of infarction and functional outcomes. The results showed that the STA sign was associated with clinical outcome in patients with CVS after SAH aged ≥50 years, and was associated with the occurrence of leptomeningeal collaterals. Moreover, the authors provide the predictive values of the presence or absence of the sign for the outcome.

The paper is innovative and draws conclusions based on the research data, and the research design is appropriate, which is adequately described in methods. The paper provides valuable DSA predictive signs for The occurrence of CVS after SAH, so that clinicians can more calmly cope with brain tissue ischemia or infarction after SAH.

This is a topic of interest to the researchers in the related areas but the paper needs some improvement before acceptance for publication. My detailed comments are as follows:

Point 1: Abbreviations and acronyms should be defined within the text upon first appearance. Such as, in page 2, Introduction, line 56, ICA would be declared at first.

Answer: The manuscript was revised. Abbreviations were defined upon first appearance.

Point 2: Please unify the terms in the paper, such as DCI, DCI-related infarction, and cerebral infarction.

Answer: The terms were unified to “DCI-related infarction”.

Point 3: In page 5, Materials and Methods 2.6, Please determine whether it is necessary to state how to distinguish STA from Supraorbital artery.

Answer: The description of the STA sign was complemented by:

The supraorbital artery is the smaller one, which runs laterally and is often overlayed in p.a. projections. In lateral projections, it crosses the protruding part of the eyebrow (Fig. 2D).”

Point 4: Please expand the significance of this study for clinical work and research in Discussion.

Answer: The discussion was expanded as follows (italic):

“The STA sign revealed a significant positive and negative predictive value for patient’s outcome in the group ≥ 50 years. Patients at risk for unfavorable outcomes (mRS 4-6) were aged < 50 years, suffered from high-grade vasospasm and showed a positive STA sign or were aged ≥ 50 years and showed a positive STA sign with indifferent vasospasm grades. The STA sign indicates self-sufficient older patients at risk for unfavorable outcome who may profit from intensified care and monitoring after SAH. Patients suffering from severe SAH are frequently in a poor neurological condition and the diagnosis of DCI according to Vergouwen et al. [22] is often challenging with a broad inter-rater variance [23]. The STA sign might be a helpful diagnostic tool.

The STA is easily accessible for extracranial color-coded duplex sonography. In this method, the positive STA sign can also be detected with bedside-monitoring showing increase antegrade flow velocities for the affected side (Figure 7). Especially in patients aged ≥ 50 years, the transtemporal color-coded duplex sonography often fails due to a missing ultrasound window. This finding needs to be further evaluated. It represents a low-cost and non-invasive method to obtain hemodynamic information in real-time [5], even in patients without a temporal ultrasound window.”

  1. Ginsberg, M.D.; Greenwood, S.A.; Goldberg, H.I. Noninvasive diagnosis of extracranial cerebrovascular disease: oculoplethysmography-phonoangiography and directional Doppler ultrasonography. Neurology 1979, 29, 623-631, doi:10.1212/wnl.29.5.623.
  2. Vergouwen, M.D.; Vermeulen, M.; van Gijn, J.; Rinkel, G.J.; Wijdicks, E.F.; Muizelaar, J.P.; Mendelow, A.D.; Juvela, S.; Yonas, H.; Terbrugge, K.G.; et al. Definition of delayed cerebral ischemia after aneurysmal subarachnoid hemorrhage as an outcome event in clinical trials and observational studies: proposal of a multidisciplinary research group. Stroke 2010, 41, 2391-2395, doi:10.1161/STROKEAHA.110.589275.
  3. Tjerkstra, M.A.; Verbaan, D.; Coert, B.A.; Post, R.; van den Berg, R.; Coutinho, J.M.; Horn, J.; Vandertop, W.P. Large practice variations in diagnosis and treatment of delayed cerebral ischemia after subarachnoid hemorrhage. World Neurosurg 2022, doi:10.1016/j.wneu.2022.01.033.

Therefore, our Overall Recommendation is: Accept after minor revision.

Answer: We like to thank you very much for reviewing our manuscript and for your comments.
